# An Assessment of the Effectiveness and Safety of Chimeric Antigen Receptor T-Cell Therapy in Multiple Myeloma Patients with Relapsed or Refractory Disease: A Systematic Review and Meta-Analysis

**DOI:** 10.3390/ijms25094996

**Published:** 2024-05-03

**Authors:** Rita Pereira, Rui Bergantim

**Affiliations:** 1Faculty of Medicine, University of Porto, 4200-319 Porto, Portugal; up201807375@edu.med.up.pt; 2i3S—Instituto de Investigação e Inovação em Saúde, University of Porto, 4200-135 Porto, Portugal; 3Cancer Drug Resistance Group, IPATIMUP—Institute of Molecular Pathology and Immunology of the University of Porto, 4200-135 Porto, Portugal; 4Clinical Hematology Department, Hospital Center of São João, 4200-319 Porto, Portugal; 5Clinical Hematology Department, FMUP—Faculty of Medicine of the University of Porto, 4200-319 Porto, Portugal

**Keywords:** multiple myeloma, relapsed/refractory, CAR-T cell therapy, adoptive immunotherapy, systematic review

## Abstract

Multiple myeloma (MM), the second most common hematologic malignancy, remains incurable, and its incidence is rising. Chimeric Antigen Receptor T-cell (CAR-T cell) therapy has emerged as a novel treatment, with the potential to improve the survival and quality of life of patients with relapsed/refractory multiple myeloma (rrMM). In this systematic review and meta-analysis, conducted in accordance with PRISMA guidelines, we aim to provide a concise overview of the latest developments in CAR-T therapy, assess their potential implications for clinical practice, and evaluate their efficacy and safety outcomes based on the most up-to-date evidence. A literature search conducted from 1 January 2019 to 12 July 2023 on Medline/PubMed, Scopus, and Web of Science identified 2273 articles, of which 29 fulfilled the specified criteria for inclusion. Our results offer robust evidence supporting CAR-T cell therapy’s efficacy in rrMM patients, with an encouraging 83.21% overall response rate (ORR). A generally safe profile was observed, with grade ≥ 3 cytokine release syndrome (CRS) at 7.12% and grade ≥ 3 neurotoxicity at 1.37%. A subgroup analysis revealed a significantly increased ORR in patients with fewer antimyeloma regimens, while grade ≥ 3 CRS was more common in those with a higher proportion of high-risk cytogenetics and prior exposure to BCMA therapy.

## 1. Introduction

Multiple myeloma (MM), whose inaugural case was first documented in the 1840s [1], stands as the second most common hematologic malignancy, comprising roughly 10% of hematologic malignancies and about 1% of all cancers globally. Notably, it is experiencing an increase in incidence and prevalence in the developed world [2,3,4].

MM is characterized by the presence of clonal plasma cells that accumulate within the bone marrow, disrupting the normal hematopoiesis, and it is further distinguished by the abnormal production of immunoglobulins, which can be detected in both serum and urine [5,6]. Upon diagnosis of MM, patients often present persistent and nonspecific symptoms, which may delay the diagnosis and early treatment initiation. However, the most frequent manifestations involve fatigue associated with anemia, bone-related pain attributed to lytic lesions and pathological fractures, edema resulting from renal failure, as well as stupor and behavioral changes due to hypercalcemia [7,8].

In recent years, significant therapeutic advancements have been made, with the advent of immunomodulatory drugs, proteasome inhibitors, and monoclonal antibodies—especially notable is the potential efficacy of combining these classes in different settings, leading to a substantial extension of the overall survival of MM patients. Importantly, the consolidation of responses with autologous stem cell transplantation (ASCT) in patients who are eligible for that continues to be a crucial component of the standard of care of these patients. Nevertheless, MM remains an incurable hematological malignancy, marked by multiple remission stages and subsequent relapses throughout its clinical course. This is attributed to its intra-, spatial-, and temporal heterogeneity and the ongoing clonal evolution at each stage, conferring MM with high-risk characteristics and resistance to standard treatments [9,10].

Genetically engineered T-cells stand out as a promising and robust emerging therapy, inspiring hope for achieving a cure in patients dealing with cancer. The aim of Chimeric Antigen Receptor T-cell (CAR-T cell) therapy is to remodel the patient’s T-cells to selectively target cancer cells. T-cells from the host are harvested and undergo genetic modifications ex vivo to exhibit a chimeric antigen receptor (CAR) designed to bind to a tumor-specific antigen. Once the T-cells are engineered to express the CAR, they are infused back into the patient, where they can target and destroy the cancer cells that express the targeted antigen. The CAR structure typically comprises an external component featuring a single-chain variable fragment (scFv) targeting the desired antigen, complemented with an internal component that anchors the structure within the T-cell membrane and triggers an internal signaling sequence [11,12]. Nonetheless, the CAR’s binding domain can originate from diverse elements. For instance, ciltacabtagene autoleucel, which the targets B cell maturation antigen (BCMA), utilizes a single variable domain on a heavy chain (VHH), also known as nanobodies, making it unique among Food and Drug Administration (FDA)-approved therapies [13]. The intracellular domain has been the central focus of most modifications in CAR-T cell therapies since their inception in the late 1980s, emphasizing its crucial role in initiating and expanding CAR-T cells [14]. Initially, first-generation CARs, which incorporated only a CD3 ζ-chain or FcεRIγ intracellular domain without additional co-stimulatory domains, were found to be ineffective in generating potent antitumor responses [15]. Subsequent advancements led to the development of second-generation CARs, which are the most used ones in multiple myeloma, featuring a single co-stimulatory domain, and third-generation CARs, combining multiple co-stimulatory domains. Our systematic review will specifically address second- and third-generation CARs. These advanced CARs have demonstrated a significant increase in potency, characterized by a notable enhancement in cytokine generation, improved efficacy against tumors, and enhanced T-cell proliferation. This evolution in design underscores the significance of incorporating co-stimulatory domains, including CD28, 4-1BB, or OX40, for optimal therapeutic outcomes [16,17,18].

BCMA, a member of the tumor necrosis factor receptor superfamily 17 (TNFRSF17) and a transmembrane glycoprotein, stands as the key surface antigen target for CAR-T cell therapy for the treatment of multiple myeloma. BCMA exhibits selective expression in malignant plasma cells, with lower levels in normal plasma cells, and it is notably absent in non-hematological tissues [6,10,19]. However, due to the fluctuating levels of BCMA expression in malignant plasma cells, BCMA downregulation, and the heterogeneous nature of tumor antigens in MM, researchers are investigating additional target antigens, including CD19, SLAMF7, GPRC5D, CD138, CD38, CD70, NKG2DL, and kappa/lambda light chains [10].

Nevertheless, the utilization of this innovative therapy is hindered by the risk of severe toxic reactions, with cytokine release syndrome (CRS) being the most common, along with additional effects such as neurotoxicity, cytopenias, and a susceptibility to infections. Therefore, supportive care is imperative for every patient experiencing these toxicities, demanding prompt intervention to maintain the benefits of CAR-T cell therapy while mitigating potentially life-threatening adverse reactions [20,21].

Over the past five years, there has been a significant surge in clinical trials and the publication of real-world data, all with the purpose of thoroughly investigating the benefits and risks of CAR-T therapy. Our goal is to conduct a comprehensive assessment of the effectiveness and safety of innovative CAR-T cell therapy in relapsed/refractory multiple myeloma patients (rrMM) through a systematic review and meta-analysis based on the most up-to-date evidence, with the ultimate goal of guiding clinicians’ decisions and enhancing clinical recommendations.

## 2. Materials and Methods

A systematic review with meta-analysis was structured according to the Preferred Reporting Items for Systematic Reviews and Meta-Analyses (PRISMA) guidelines [22].

### 2.1. Eligibility Criteria

Studies from 1 January 2019 to 12 July 2023 were considered for analysis in this review if they fulfilled the specified criteria, in accordance with the PICO framework [23]: (1) Study types: clinical trials and cohort studies, either prospective or retrospective; (2) Population: patients (aged 18 years or older) with relapsed/refractory multiple myeloma; (3) Intervention: CAR-T cell therapy, regardless of the specific antigen targeted; (4) Outcomes: minimum of one efficacy and one safety assessment. Efficacy outcomes included overall response rate (ORR), complete response rate (CRR), very good partial response (vgPR), partial response (PR), progressive disease (PD), measurable residual disease (MRD), negativity in the stringent complete response (sCR)/complete response (CR) group, median progression-free survival (mPFS), and median duration of response (mDOR). The CRR integrates the sCR and CR. The ORR, defined according to International Myeloma Working Group (IMWG) response criteria [24], represents the proportion of patients achieving a PR or better. Safety outcomes encompassed CRS, neurotoxicity, hematology-related adverse events (neutropenia, leukopenia, anemia, thrombocytopenia, lymphopenia), infections, and all-cause mortality. Except for all-cause mortality, all remaining safety outcomes were assessed for any grade and for grade ≥ 3. The most commonly used criteria for grading the severity of adverse events in safety outcomes were the National Cancer Institute Common Terminology Criteria for Adverse Events (NCI CTCAE) versions 4.03 and 5.0, as well as the criteria published by Lee et al. [25,26].

Studies were excluded from the analysis if they met the following criteria: (1) review articles, abstracts, conference presentations, case–control studies, case reports, letters to the editor, and editorials; (2) language other than English; and (3) studies conducted in non-human subjects. In cases of similar and repeated clinical trials, only the one with the longest follow-up time was included for analysis. This review was limited to articles with full-text availability, and in cases where it was absent, the authors were contacted.

### 2.2. Information Sources and Search Strategy

Relevant articles were searched in the three indexed literature databases, Medline/PubMed, Scopus, and Web of Science. The last search was conducted with the last survey performed on 12 July 2023. A time limit was set from 1 January 2019 to 12 July 2023, and no language restrictions were applied in this phase.

The PubMed search query was ((“multiple myeloma” [MeSH Terms] OR “multiple myeloma”[Title/Abstract] OR “relapsed multiple myeloma”[Title/Abstract] OR “refractory multiple myeloma”[Title/Abstract]) AND (“immunotherapy, adoptive”[MeSH Terms] OR “car t cell”[Title/Abstract] OR “chimeric antigen receptor T-cell therapy”[Title/Abstract] OR “car t immunotherapy”[Title/Abstract])). Searches in additional databases were adjusted using this query as a reference. Appendix A provides a detailed overview of the search strategy.

### 2.3. Study Selection Process

After gathering all the articles, duplicates were identified and removed using the Rayyan online platform. Initially, studies were assessed based on the title and abstract by two independent reviewers. Eligible articles underwent a full-text independent evaluation following the outlined criteria, with any discordances resolved through consensus.

### 2.4. Data Extraction

Data were manually collected by two independent reviewers using a pre-defined form. Once more, a consensus methodology was employed to resolve any disagreements.

Data were collected regarding the following: (1) study characteristics: first author, publication year, study design, production name, registration number, study phase, country, sample size, bridging therapy, lymphodepletion regimen, CAR-T cell dose, median follow-up time, target antigen, T-cell source, gene transfer method, co-stimulatory domain, scFv (single-chain variable fragment) species, and CAR-T cell generation; (2) patients: age, gender, race, median time since diagnosis, ECOG PS score, ISS staging, extramedullary disease, high-risk cytogenetic profile, median number of previous antimyeloma regimens, prior treatment class, previous CAR-T cell therapy, previous BCMA therapy, and ASCT before CAR-T; (3) evaluated outcomes.

### 2.5. Risk of Bias

The quality assessment of non-randomized studies was conducted by two independent reviewers using the Methodological Index for Non-Randomized Studies (MINORS) [27]. The evaluated criteria consisted of 8 elements: clearly stated aim, inclusion of consecutive patients, prospective collection of data, endpoints appropriate to the aim of the study, unbiased assessment of the study endpoint, follow-up period appropriate to the aim of the study, loss to follow-up less than 5%, and prospective calculation of the study size. Each study could attain a maximum score of 16. A valuation of 0 (not reported), 1 (reported but inadequate), or 2 (reported and adequate) was attributed for each criterion to assess the quality of the study.

### 2.6. Data Synthesis and Statistical Analysis

Statistical analyses to evaluate studies assessing the impact of CAR-T cell therapy on patients with rrMM were performed using the R studio software 2023.06.2+561 (meta-package).

The following outcomes were selected for inclusion in the meta-analysis: ORR, CRR, vgPR, PR, PD, MRD negativity in the sCR/CR group, mPFS, mDOR, CRS, neurotoxicity, blood-related adverse events (neutropenia, leukopenia, anemia, thrombocytopenia, lymphopenia), infections, and all-cause mortality.

Proportions and weighted medians were employed to measure the effects and present them with their respective 95% confidence intervals (95% CIs). To investigate heterogeneity among studies, Cochran’s Q test and I^2^ statistic were employed, considering significant heterogeneity when *p* < 0.10 and/or I^2^ > 40%. The random effects model was adopted in these circumstances. Subgroup analysis was conducted to identify potential sources of heterogeneity, considering factors such as the number of prior antimyeloma regimens (<5 vs. ≥5), previous exposure to BCMA therapy (yes vs. no), previous ASCT (<78% vs. ≥78%), high-risk cytogenetics (<39% vs. ≥39%), ISS stage 3 (<24% vs. ≥24%), extramedullary disease (<28% vs. ≥28%), bridging therapy (<42% vs. ≥42%), CAR-T generation (second vs. third), and upper infusion threshold (<490 × 10^6^ cells or 2.05 × 10^6^ cells/Kg vs. ≥490 × 10^6^ cells or 2.05 × 10^6^ cells/Kg). A *p*-value < 0.05 was considered statistically significant. Forest plots were used to illustrate the analysis results, and the presence of publication biases was examined through funnel plots. Additionally, for a quantitative assessment of publication biases, Egger´s test and Begg´s test were employed, with a *p*-value greater than 0.05 indicating no significant publication biases.

## 3. Results

### 3.1. Study Selection

The initial search on the PubMed, Scopus, and Web of Science literature databases identified 705, 974, and 594 articles, respectively, resulting in a combined total of 2273 articles that could potentially hold importance. After eliminating 1049 duplicates, the remaining 1224 were screened by title and abstract, with 52 proceeding to full-text assessment for eligibility. Out of those, 29 articles were eligible to be included in this review [28,29,30,31,32,33,34,35,36,37,38,39,40,41,42,43,44,45,46,47,48,49,50,51,52,53,54,55,56]. The PRISMA flow diagram illustrating the study selection process is presented in Figure 1.

### 3.2. Study Characteristics and Initial Qualities of the Enrolled Patients

Twenty-nine articles published between 2019 and 2023 were included in this review. These included 27 prospective studies and 2 retrospective studies. Among these, 2 assessed real-world data, 15 were reported as phase I clinical trials, 3 were phase I/II trials, and 9 were phase II trials. Geographically, seven studies were conducted exclusively in the USA, two in Japan, one in Switzerland, one in Spain, one in Israel, and the majority, fourteen, were conducted exclusively in China. Additionally, three studies were conducted across multiple countries. The collective cohort included 1051 multiple myeloma patients who underwent CAR-T treatment. Predominantly, BCMA was investigated as the main target (19 studies, 66%). However, dual targeting and GPRC5D were also explored in seven and three articles, respectively. Except for studies conducted by Yan et al. (2020) [31], Chen et al. (2020) [32], Wang et al. (2022) [35], and Lee et al. (2023) [49], which evaluated third-generation CAR-T therapy, all the other studies utilized second-generation CAR-T therapies. Details of the CAR-T constructs are provided in Appendix A. The predominant lymphodepleting conditioning regimen used was cyclophosphamide plus fludarabine. The doses of CAR-T cells varied among studies, and follow-up periods spanned from 136 days to 48 months. A comprehensive summary of the characteristics of the included articles is detailed in Appendix A and Table 1.

All studies included in the analysis reported participant ages ranging from 27 to 83 years. Data regarding gender were reported in 28 studies, with a male proportion of 59% (613/1038). Information on race was only present in six studies. According to the International Staging System (ISS), 25% (246/985) of patients were classified as stage III, as reported in 25 articles. The mean number of previous antimyeloma regimens across all studies was five. Extramedullary disease was observed in 30% (302/1010) of the patients, as documented in 26 articles. Additionally, 4% (30/805) of patients had received prior CAR-T cell therapy, 9.45% (93/984) had undergone previous BCMA therapy, and 66% (697/1054) had undergone autologous stem cell transplant (ASCT) before CAR-T, as reported in 24, 26, and 29 articles, respectively. Specifics regarding each patient´s multiple myeloma condition are outlined in Appendix A and Table 2. 

### 3.3. Meta-Analysis Results

#### 3.3.1. Efficacy Outcomes

ORR: All included studies, comprising a total of 1047 patients, reported results for ORRs. Among these, 872 patients achieved an overall response, resulting in a cumulative incidence rate of 83.21% (95% CI, 75.50–89.85; I^2^ = 83%; Figure 2).

CRR, vgPR, PR, and PD: The meta-analysis revealed pooled rates of 50.31% (95% CI, 40.47–60.13; I^2^ = 86%; Appendix A), 16.38% (95% CI, 12.92–20.12; I^2^ = 48%; Appendix A), 8.74% (95% CI, 5.47–12.54; I^2^ = 66%; Appendix A), and 3.70% (95% CI, 1.09–7.30; I^2^ = 69%; Appendix A) for the CRR, vgPR, PR, and PD, respectively.

MRD negativity in the sCR/CR group: Among the 15 studies that evaluated measurable residual disease in the sCR/CR group, it was estimated to be 84.51% (95% CI, 73.39–93.47; I^2^ = 84%; Figure 3).

mPFS and mDOR: The mPFS was 8.63 months (95% CI, 4.76–12.49; I^2^ = 92%; Appendix A) across eight included studies. In the case of the mDOR, the value was 14.48 months (95% CI, 8.73–20.23; I^2^ = 59%; Figure 4) based on five studies. The limited inclusion of studies stems from a significant portion lacking available data on the 95% confidence interval and/or failing to achieve the upper limit.

#### 3.3.2. Safety Outcomes

CRS: In relation to safety, all 1051 patients were included for evaluation of any grade and grade ≥ 3 CRS rates. The proportion of CRS of any grade was 85.89% (95% CI, 78.67–91.98; I^2^ = 84%; Appendix A), and for grade ≥ 3, it was 7.12% (95% CI, 3.58–11.49; I^2^ = 72; Figure 5).

*Neurotoxicity:* The estimate of any grade of neurotoxicity was 8.27% (95% CI, 4.43–12.93; I^2^ = 73%; Appendix A), and for grade ≥ 3 neurotoxicity, it was 1.37% (95% CI, 0.27–3.03; I^2^ = 28%; Figure 6).

Hematology-related adverse events: The combined effect size demonstrated that the proportions of any grade of neutropenia, leukopenia, anemia, thrombocytopenia, and lymphopenia were 95.93% (95% CI, 91.99–98.79; I2 = 74%; Appendix A), 84.76% (95% CI, 70.99–95.05; I^2^ = 95%; Appendix A), 81.09% (95% CI, 71.42–89.31; I^2^ = 91%; Appendix A), 74.49% (95% CI, 64.54–83.40; I^2^ = 90%; Appendix A), and 65.54% (95% CI, 42.23–85.71; I^2^ = 96%; Appendix A), respectively. Furthermore, the pooled proportions of patients experiencing grade ≥ 3 hematology-related adverse events were 88.02% (95% CI, 80.38–94.18; I^2^ = 83%; Appendix A) for neutropenia, 72.75% (95% CI, 59.14–84.66; I^2^ = 93%; Appendix A) for leukopenia, 48.16% (95% CI, 39.88–56.49; I^2^ = 82%; Appendix A) for anemia, 48.98% (95% CI, 40.14–57.86; I^2^ = 83%; Appendix A) for thrombocytopenia, and 63.91% (95% CI, 41.33–83.86; I^2^ = 96%; Appendix A) for lymphopenia.

Infections: Based on data from 17 studies, the overall proportion of any grade of infection was 45.36% (95% CI, 33.89–57.05; I^2^ = 86%; Figure 7), and the proportion was 17.52% (95% CI, 7.91–29.43; I^2^ = 81%; Figure 8) for ≥grade 3 infections among 11 included studies.

All-cause mortality: Data regarding mortality from all causes were outlined in 28 studies, resulting in a cumulative mortality rate of 23.34% (95% CI, 16.57–30.79; I^2^ = 78%; Appendix A).

#### 3.3.3. Subgroup Analysis

To explore potential differences in the safety and efficacy of the CAR-T cell therapy under investigation, subgroup analyses were carried out. Factors considered included the number of prior antimyeloma regimens (<5 vs. ≥5), prior exposure to BCMA therapy (yes vs. no), the history of previous ASCT (<78% vs. ≥78%), the presence of high-risk cytogenetics (<39% vs. ≥39%), the proportion of patients in ISS stage 3 (<24% vs. ≥24%), the presence of extramedullary disease (<28% vs. ≥28%), the use of bridging therapy (<42% vs. ≥42%), the CAR-T generation (second vs. third), and the upper infusion threshold (<490 × 10^6^ cells or 2.05 × 10^6^ cells/Kg vs. ≥490 × 10^6^ cells or 2.05 × 10^6^ cells/Kg). A comprehensive summary of these results is provided in Table 3.

Regarding the ORR, a significantly higher ORR was observed in patients with a history of <5 prior antimyeloma regimens compared to those who had received ≥ 5 prior antimyeloma regimens (89% vs. 75%, *p* < 0.001). Nevertheless, the subgroup analysis for the ORR based on other factors did not reveal any significant differences.

In relation to CRS of grade 3 or higher, significantly higher proportions were observed in specific patient groups. Patients with prior exposure to BCMA therapy exhibited a higher rate compared to those without previous exposure (13% vs. 5%, *p* = 0.04). Similarly, patients with prior ASCT < 78% experienced a significantly elevated occurrence of grade ≥ 3 CRS in comparison to those who had received prior ASCT ≥ 78% (15% vs. 6%, *p* = 0.02). Additionally, patients with high-risk cytogenetics ≥ 39% demonstrated a notably higher rate of grade ≥ 3 CRS compared to those with high-risk cytogenetics < 39% (15% vs. 7%, *p* = 0.02). However, the differences in grade ≥ 3 CRS in the other subgroup analyses did not reach statistical significance. The forest plots of each subgroup analysis are illustrated in Figure 9, Figure 10, Figure 11 and Figure 12 and Appendix A.

### 3.4. Risk of Bias in the Included Studies

The assessment of study quality using the MINORS score revealed a high quality in the included research, with an average score of 13 and individual scores ranging from 11 to 16. The detailed scores for the 29 included articles are presented in Appendix A.

We examined the potential for biases through a visual analysis of the funnel plot for the ORR. As demonstrated in Figure 13, the non-symmetrical distribution on both sides of the funnel plot indicates potential biases. Subsequently, the Egger and Begg tests were conducted, yielding non-significant results (*p* = 0.126 and *p* = 0.348, respectively). These results, suggesting a reduced likelihood of publication biases, also highlight that heterogeneity could potentially contribute to the observed asymmetry.

## 4. Discussion

Over the past few years, the paradigm of multiple myeloma treatment has undergone a transformative shift, moving from primarily relying on chemotherapy-based protocols to embracing immunotherapy. CAR-T cell therapy has emerged as a groundbreaking modality of treatment, revolutionizing the therapeutic algorithm for patients with rrMM. This transformation is exemplified by the approval of two anti-BCMA CAR-T therapies: idecabtagene vicleucel (ide-cel) and ciltacabtagene autoleucel (cilta-cel) in March 2021 and March 2022, respectively [57]. In fact, in a pivotal phase 3 randomized clinical trial conducted by Rodriguez-Otero et al. [58], ide-cel was compared to standard regimens (daratumumab, pomalidomide, and dexamethasone; daratumumab, bortezomib, and dexamethasone; ixazomib, lenalidomide, and dexamethasone; carfizomib and dexamethasone; or elotuzumab, pomalidomide, and dexamethasone) in patients with rrMM who had been exposed to triple-class agents. The results of this trial demonstrated superior and deeper treatment responses with ide-cel, as well as a safety profile that was similar to that reported by Munshi et al. [46] and Raje et al. [29]. In the ide-cel group, the mPFS reached a remarkable 13.3 months (vs. 4.4 months in the standard-regimen group, HR 0.49). Moreover, with an 18.6-month median follow-up, 71% of patients treated with ide-cel achieved a PR or better (vs. 42%), and 39% achieved a CRR (vs. 5%) (*p* < 0.001). Of the 225 patients receiving ide-cel, 88% experienced CRS, with 5% of the cases classified as grade 3 or higher. Additionally, 15% of patients had neurotoxic effects, with 3% experiencing grade 3 or higher.

The findings of our systematic review and meta-analysis reveal the efficacy and safety of CAR-T cell therapy in the context of rrMM. Spanning the years 2019 to 2023 and incorporating 29 articles, with a collective cohort of 1051 multiple myeloma patients, it offers a comprehensive overview of the impact of CAR-T cell on rrMM, considering various factors such as prior treatments, disease characteristics, and treatment-related adverse effects. Most of the included articles were non-randomized controlled trials with a high level of quality, as assessed by the MINORS score.

We found that CAR-T cell treatment was remarkably effective in this challenging patient population, with an encouraging ORR of 83.21%. This included a CRR of 50.31%, a vgPR of 16.38%, and a PR of 8.74%. The proportion of PD cases was relatively low, calculated at 3.70%. Notably, the subgroup analysis showed a significantly higher ORR in patients with a history of fewer than five prior antimyeloma lines of treatment compared to those with five or more (89% vs. 75%, *p* < 0.001). This observation aligns with previous studies suggesting that starting new approaches early in the treatment course and having a lower pre-CART cell burden of disease are associated with a more favorable treatment response, emphasizing the need for a better strategic treatment planning [59].

As evidence, in a randomized phase 3 clinical trial reported by San-Miguel et al. [60], cilta-cel was compared to standard regimens (pomalidomide, bortezomib, and dexamethasone; daratumumab, pomalidomide, and dexamethasone) in patients with rrMM who were refractory to lenalidomide after one to three prior lines of therapy. Compared to the study reported by Rodriguez-Otero et al. [58], this research specifically targeted a population in an earlier stage, with only 23% being CD38-refractory (vs. 95%), 14% being triple class-refractory (vs. 66%), and 73% having received only one or two prior lines of therapy (vs. three). In the cilta-cel group, with a median follow-up of 15.9 months, the ORR was 85% (vs. 67% in the standard-regimen group), the CR rate was 73% (vs. 22%), and the MRD-negative rate (at 10^5^) was 61% (vs. 16%). Of the 176 patients receiving cilta-cel, 76% experienced CRS, with 1% of the cases being classified as grade 3 or higher. Additionally, 4.5% of patients had ICANS, all grade 1 or 2.

Despite promising results emerging from recent research, further studies are needed to explore the potential of frontline CAR-T cell therapy. Fortunately, there are ongoing clinical trials evaluating CAR-T cell therapy for patients with newly diagnosed multiple myeloma (NDMM). For example, the CARTITUDE-6 [61] is a randomized phase 3 study aiming to compare the efficacy of daratumumab, bortezomib, lenalidomide, and dexamethasone (DVRd) followed by cilta-cel and lenalidomide versus DVRd followed by ASCT, DVRd, and lenalidomide. Also, in cases where ASCT is not intended to be the initial therapeutic approach, the CARTITUDE-5 study [62], a randomized phase 3 trial will compare the efficacy of Bortezomib, Lenalidomide, and Dexamethasone (VRd) induction followed by cilta-cel versus VRd induction followed by Lenalidomide and Dexamethasone (Rd) maintenance therapy. The results of these ongoing studies are eagerly awaited to evaluate the true potential of CAR-T therapy in the initial treatment of multiple myeloma [63].

Furthermore, in our meta-analysis, fifteen studies provided data on MRD negativity among the sCR/CR group. MRD negativity was estimated at 84.51%, which is indicative of a substantial proportion of patients achieving a deep response. Munshi et al. [64] reenforce that assessing MRD serves as a predictive factor in multiple myeloma, as achieving deeper responses is associated with favorable survival outcomes, namely, prolonged PFS, irrespective of the MRD detection method employed. Therefore, in our meta-analysis, the mPFS and the mDOR were 8.63 and 14.48 months, respectively.

Nonetheless, the application of CAR-T cell therapy can be limited by its side effects. CRS, the most common CAR-T-related toxicity, is a systemic inflammatory response that arises from immune activation associated with the proliferation of CAR-T cells. It releases a large number of cytokines, especially IL-6, IL-10, and interferon (IFN)-Υ, which surpasses the capacity of self-regulating homeostatic mechanisms to control the reaction [65]. CRS can initiate within a few days and typically resolves in 2 to 3 weeks, depending on factors such as patient attributes, CAR-T characteristics, and therapeutic approaches [66]. This clinical condition can vary from mild symptoms like muscle and joint pain, rash, headache, fever, and fatigue to severe presentations, including shock, blood clotting issues, fluid leakage, and organ failure. In rare cases, it may resemble macrophage activation syndrome (MAS) in both clinical and lab findings [65,67]. In our analysis, we found a substantial proportion of patients experiencing any grade of CRS, accounting for 85.89%. Importantly, only 7.12% had grade ≥ 3, particularly those with prior exposure to other BCMA therapies, a history of prior ASCT, and high-risk cytogenetics. These findings highlight the importance of monitoring and managing CRS, especially in patients with these identified risk factors.

On the other hand, the underlying pathophysiology of CAR-T-associated neurotoxicity remains uncertain. Neuroimaging studies are frequently performed, yet they seldom revealed structural abnormalities. The spectrum of this clinical condition can range from encephalopathy and mental status changes to cerebral edema, seizures, and possible death [68,69,70]. In our analysis, any grade and grade ≥ 3 of neurotoxicity were extremely low, at 8.27% and 1.37%, respectively. Supportive care, tocilizumab, and glucocorticoids constitute the primary approaches to toxicity management in most patients [69].

The presented meta-analysis revealed a considerable infection rate, with a 45.36% rate of any-grade infections and a 17.52% rate of grade 3 or higher infections. Individuals with multiple myeloma undergoing CAR-T cell therapy may experience infections at various intervals post-treatment. The factors contributing to the increased infection risk appear to be multifactorial and are attributed to the CAR-T cell treatment itself, as well as prior treatments and the underlying disease. Specifically, hypogammaglobulinemia resulting from plasma cell aplasia, along with cytopenias (neutropenia and lymphopenia) and T-cell exhaustion, have been described [71]. We believe that infections during CAR-T therapy can result in extended hospitalization, significantly impacting the patients’ quality of life. Furthermore, they may lead to treatment interruptions, altering the planned therapy and potentially compromising overall effectiveness. Therefore, effective infection management becomes crucial for achieving optimal treatment outcomes.

However, CAR-T therapy represents a recent innovation, and recommendations for managing related toxicities continue to evolve under the guidance of experts at the forefront of the field. Given the potentially life-threatening nature of CAR-T-related toxicities, it is crucial to work toward the identification of modifiable factors to minimize the risk of such adverse effects. There is emerging evidence suggesting that the dosage of CAR-T therapy may impact safety, with higher doses being associated with an increased risk of toxicities [72]. In our subgroup analysis, the ORR and an incidence of grade ≥ 3 CRS appeared slightly higher and lower, respectively, in studies with an upper infusion threshold < 490 × 106 cells or 2.05 × 106 cells/kg compared to studies with an upper infusion threshold ≥ 490 × 106 cells or 2.05 × 106 cells/kg. However, it is important to note that these differences were not statistically significant, as indicated by *p*-values of 0.71 and 0.41, respectively.

Frigault et al. [73] found that administering CAR-T cell therapy in fractions appears to have the potential to decrease the frequency and intensity of CAR-T cell toxicity while preserving the treatment’s efficacy [74]. In one of the included studies in our analysis, as reported by Oliver-Caldés et al. [50], a dose-fractionated scheme was used, complemented by an additional booster dose administered 100 days after the initial infusion. This approach demonstrated low levels of toxicity, with no reported cases of immune-effector-cell-associated neurotoxicity syndrome (ICANS) or late neurotoxicity. Moreover, there were no occurrences of grade 3 CRS or higher. The authors conducting this clinical investigation established a connection between the reduced occurrence of CAR-T-therapy-related toxicities, especially the severe ones, with the fractionation of the initial dose. This correlation was previously outlined by Ortiz-Maldonado et al. [75] in their research on ARI-0001 CART19-cell therapy for acute lymphoblastic leukemia. Despite these clinical trials beginning to provide support for the previously mentioned assertions, the supporting data for dose fractionation are not extensive, and additional research is crucial to confirm its advantages. Additionally, due to the limited number of studies included with fractionated dose regimens, we were unable to perform subgroup analysis based on CAR-T infusion timing.

Bridging therapy (BT) is recommended for most patients due to the fast progression of the disease, and it is crucial to understand how BT impacts clinical outcomes. Afrough et al. [76], through an analysis of data from eleven academic centers in the US, provide valuable information on this topic. In this real-world analysis, patients who received BT were compared to a group that did not receive bridging therapy (NBT) before undergoing treatment with ide-cel. Among the 170 patients who received BT, 35.5% were treated with an alkylator; 14% received immunomodulatory drugs, either with or without monoclonal antibodies; 12% had proteasome inhibitor combinations; and 10% were treated with Selinexor. Overall, BT was associated with a worse PFS (8.1 months in BT vs. 11.5 months in NBT; *p* = 0.03) and was also predictive of worse OS (13.8 months in BT vs. not reached in NBT; *p* = 0.002). Thus, the use of BT alongside CAR-T cell therapy might not lead to improved results in patients with rrMM, highlighting a more aggressive disease phenotype. Once again, these patients might benefit from initiating CAR-T therapy early on in their disease progression.

Also, it is crucial to be cautious when using BCMA-targeted agents before BCMA-directed CAR-T therapy, given recent data indicating less favorable outcomes in those with prior BCMA therapy exposure. In a real-world analysis conducted by Ferreri et al. [77], patients with prior exposure to BCMA-directed therapies, including antibody–drug conjugates, bispecifics, and CAR-Ts, experienced a significantly inferior OS rate (74% vs. 88%; *p* = 0.021), mDOR (7.4 vs. 9.6 months; *p* = 0.03), and mPFS (3.2 vs. 9.0 months; *p* = 0.0002) following ide-cel therapy. Moreover, according to the study by Cohen et al. [41], which is included in our analysis, a shorter duration of prior anti-BCMA treatment and an extended period between the anti-BCMA treatment and cilta-cel infusion seem to impact the responsiveness to cilta-cel. Furthermore, our analysis suggests a correlation between prior BCMA therapy exposure and the frequency of adverse events. Specifically, patients with prior BCMA exposure exhibited a significantly higher ≥ 3 CRS rate compared to those without prior exposure (13% vs. 5%, *p* = 0.004). Although findings in some studies indicate better response rates and durability in individuals without prior BCMA therapies, CAR-T therapy continues to demonstrate clinical benefits in patients with prior BCMA exposure. Despite some disagreement regarding the significance of prior BCMA therapy, our subgroup analysis of ORR based on prior BCMA exposure revealed no significant differences, suggesting that the observed effect size among studies may be relatively modest.

Despite the valuable insights provided, when interpreting the results from this systematic review and meta-analysis, it is important to acknowledge certain limitations. First and foremost, the heterogeneity among studies found in this meta-analysis must be considered. Despite conducting subgroup analysis, significant heterogeneity persisted. Heterogeneity may arise from multiple factors, including disease characteristics and patient attributes. For instance, this meta-analysis included patients from diverse ethnic groups, spanning wide ranges of ages and encompassing patients with very different criteria of certain diseases characteristics, such as cytogenetics. Also, the included studies exhibited diverse designs/methodologies and durations of follow-up. Likewise, several included studies explored different CAR-T constructs. These factors can contribute to increased heterogeneity across the studies, potentially influencing the analysis results. Second, randomized controlled trials were not included because only one was found, resulting in a lower quality of the presented evidence. Third, within the included studies, different grade criteria for CRS and neurotoxicity were employed, potentially leading to discrepancies in the reported frequency and severity of both conditions. Fourth, to estimate the pooled median overall survival (mOS), mPFS, and mDOR, many studies had no available data on the 95% confidence interval and/or did not reach the upper limit. Therefore, the estimation of mOS was not feasible, and the results of mPFS and mDOR estimation may not be as accurate as desired.

Regardless of the previously mentioned limitations, we believe that our systematic review provides valuable insights about the efficacy and safety of CAR-T cell therapy in patients with rrMM, considering the inclusion of a relatively large number of patients with global representation and the assessment of twenty-five outcomes. Additionally, to mitigate heterogeneity, a random effects model was employed, and nine subgroup analyses were conducted. Consequently, our study serves as support for more robust clinical trials that aim to elucidate the full potential and limitations of CAR-T cell therapy in this patient population.

## 5. Conclusions

In conclusion, this systematic review and meta-analysis provide evidence supporting the effectiveness of CAR-T cell therapy in the treatment of rrMM patients, with a promising ORR of 83.21%, MRD negativity in the sCR/CR group of 84.51%, and mDOR of 14.48 months. This therapy also maintains a favorable safety profile, with a proportion of grade ≥ 3 CRS events of 7.12% and grade ≥ 3 neurotoxicity of 1.37%. Additionally, it is crucial to highlight that a significantly increased ORR was observed in patients receiving a lower number of antimyeloma regimens. Moreover, grade ≥ 3 CRS was more frequent in patients with a higher proportion of high-risk cytogenetics and prior exposure to BCMA therapy. The observed influence of prior treatment history and the patient´s myeloma profile on outcomes may contribute valuable insights to enhance clinical recommendations for multiple myeloma management.

## Figures and Tables

**Figure 1 ijms-25-04996-f001:**
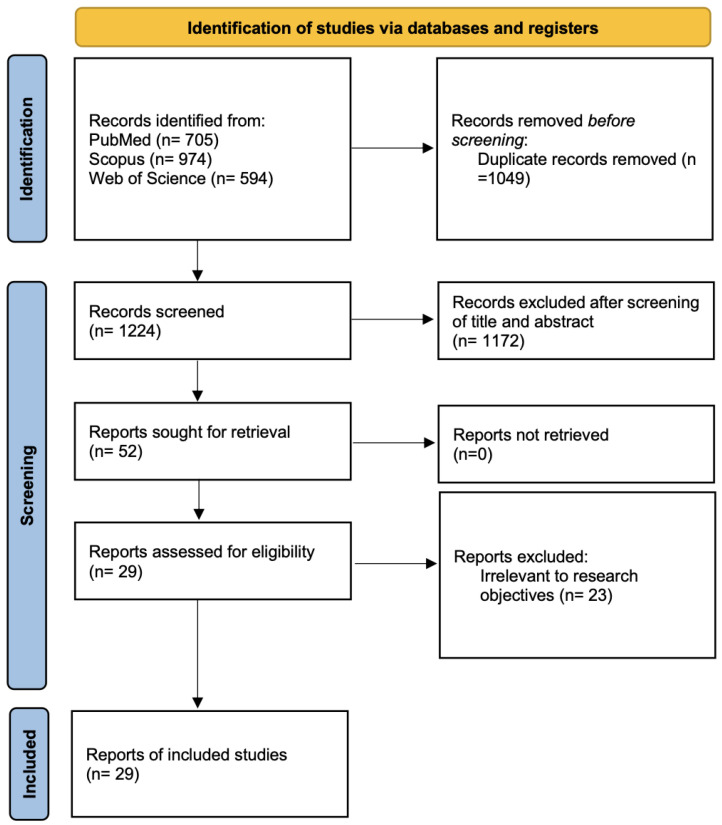
PRISMA flow diagram.

**Figure 2 ijms-25-04996-f002:**
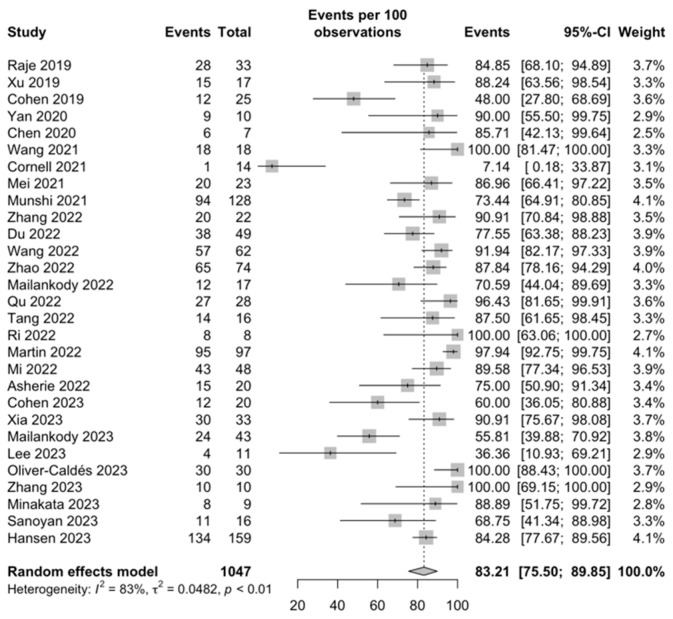
ORRs [28,29,30,31,32,33,34,35,36,37,38,39,40,41,42,43,44,45,46,47,48,49,50,51,52,53,54,55,56].

**Figure 3 ijms-25-04996-f003:**
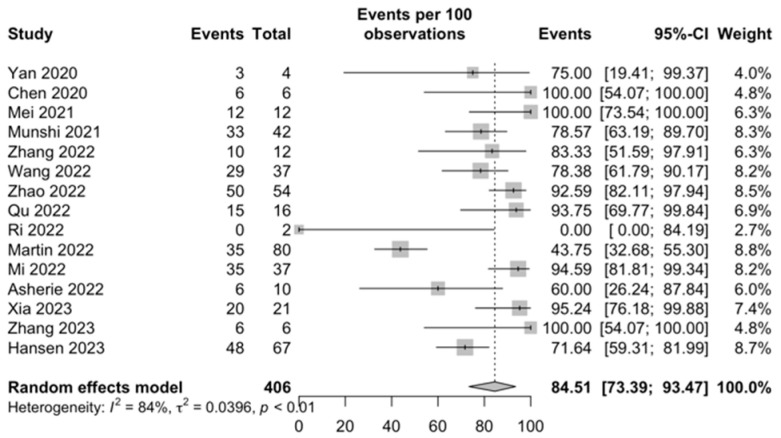
MRD negativity in the sCR/CR group [31,32,33,35,36,38,40,43,45,46,48,51,53,54,56].

**Figure 4 ijms-25-04996-f004:**
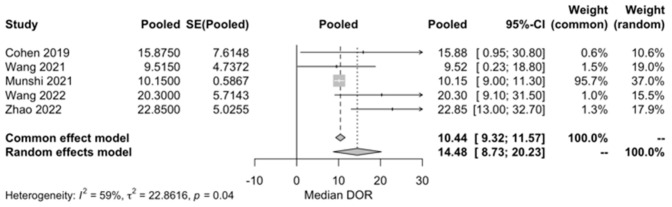
mDOR [28,35,36,46,52].

**Figure 5 ijms-25-04996-f005:**
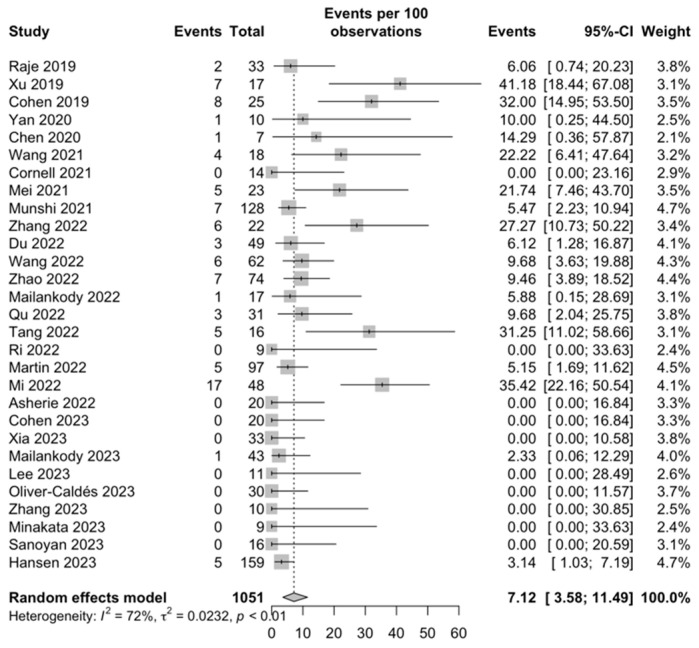
CRS grade ≥ 3 [28,29,30,31,32,33,34,35,36,37,38,39,40,41,42,43,44,45,46,47,48,49,50,51,52,53,54,55,56].

**Figure 6 ijms-25-04996-f006:**
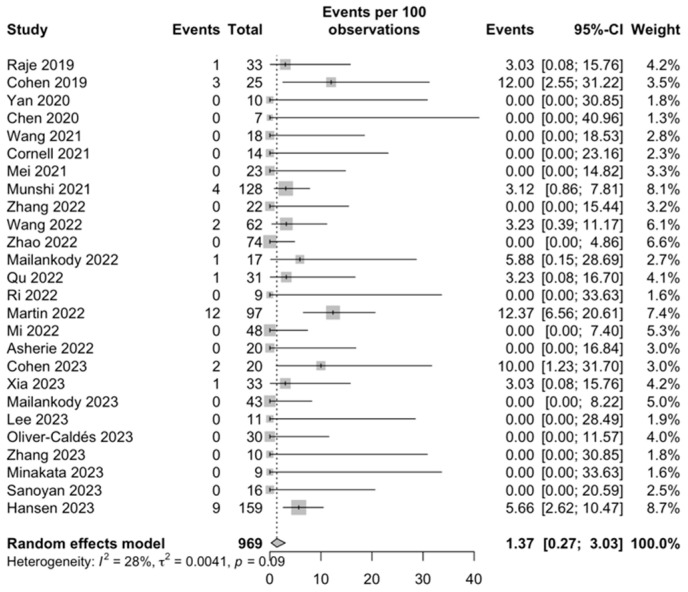
Grade ≥ 3 neurotoxicity [28,29,31,32,33,35,36,37,38,40,41,42,43,44,45,46,47,48,49,50,51,52,53,54,55,56].

**Figure 7 ijms-25-04996-f007:**
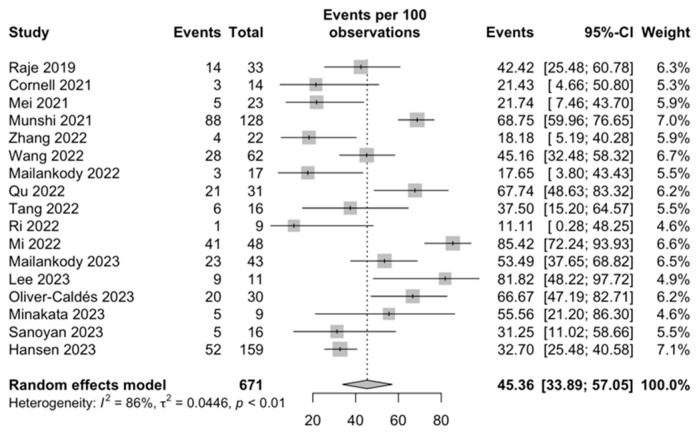
Any grade of infection [29,33,35,37,38,39,40,42,44,45,46,47,49,50,53,55,56].

**Figure 8 ijms-25-04996-f008:**
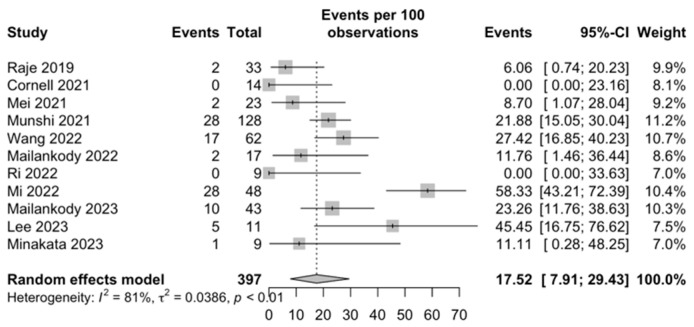
Grade ≥ 3 infections [29,35,37,40,42,44,45,46,47,49,53].

**Figure 9 ijms-25-04996-f009:**
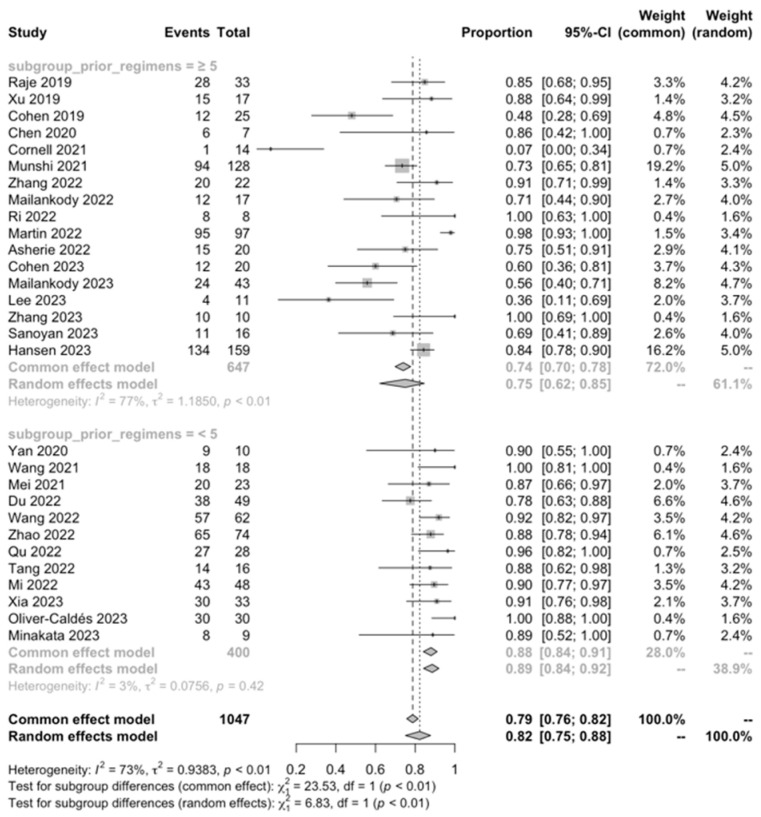
ORR, prior antimyeloma regimens [28,29,30,31,32,33,34,35,36,37,38,39,40,41,42,43,44,45,46,47,48,49,50,51,52,53,54,55,56].

**Figure 10 ijms-25-04996-f010:**
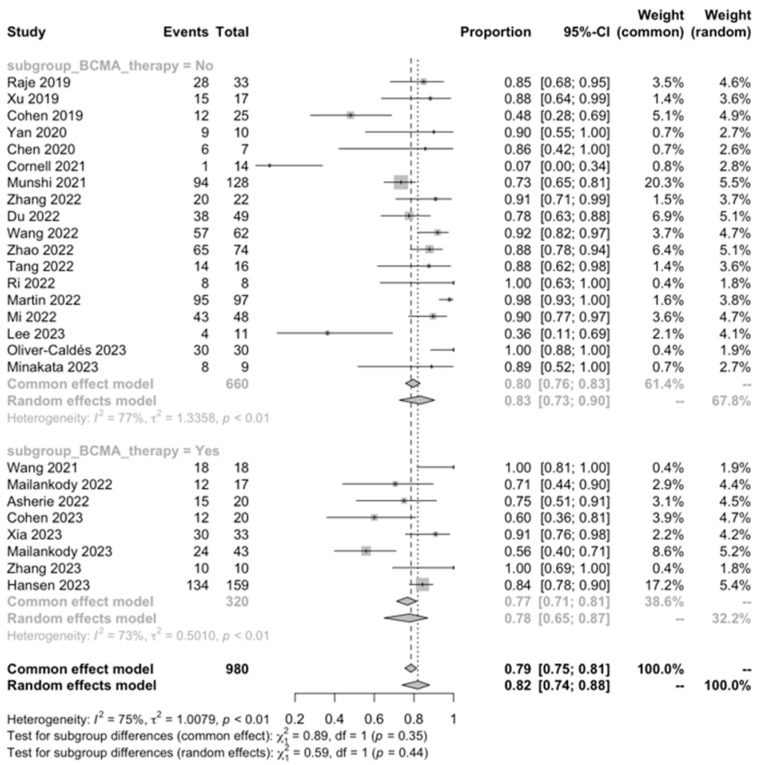
ORR, prior exposure to BCMA therapy [28,29,30,31,32,33,34,35,36,37,39,40,41,42,43,44,46,47,48,49,50,51,52,53,54,56].

**Figure 11 ijms-25-04996-f011:**
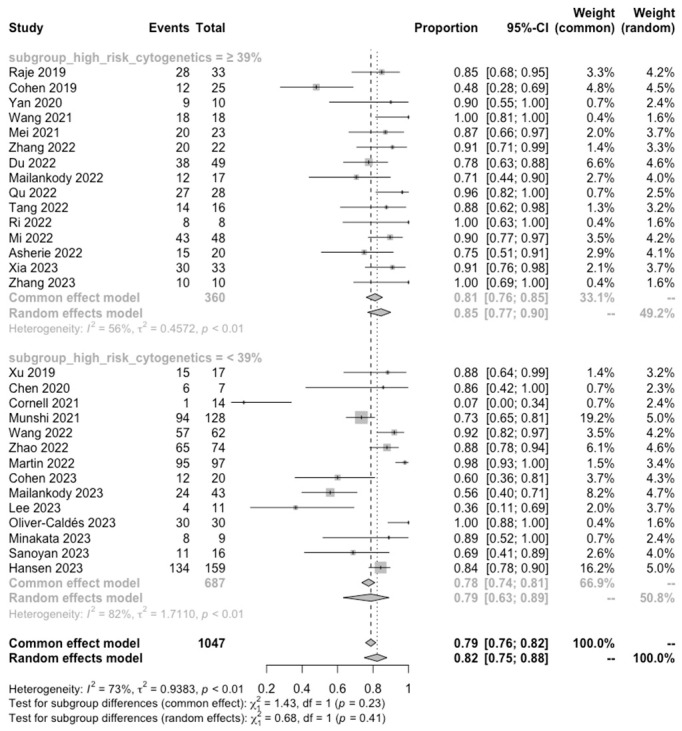
ORR, high-risk cytogenetics [28,29,30,31,32,33,34,35,36,37,38,39,40,41,42,43,44,45,46,47,48,49,50,51,52,53,54,55,56].

**Figure 12 ijms-25-04996-f012:**
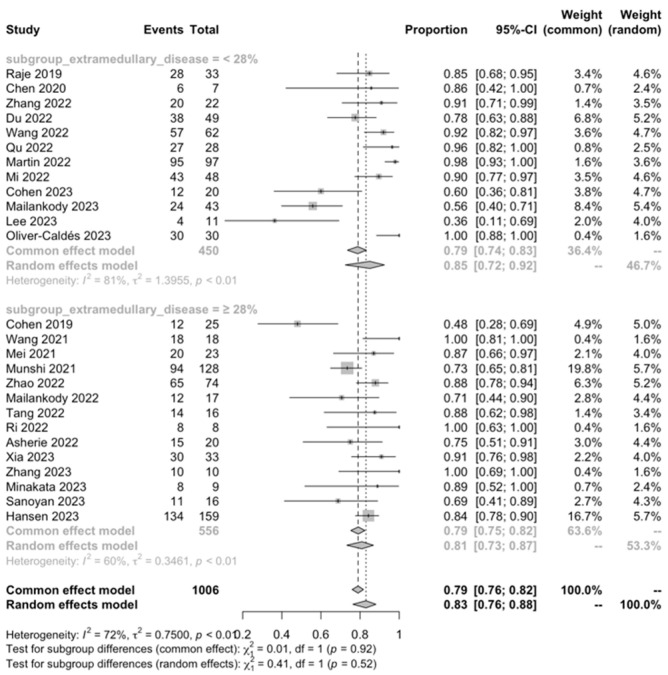
ORR, extramedullary disease [28,29,32,33,34,35,36,37,38,39,40,41,42,43,45,46,47,48,49,50,51,52,53,54,55,56].

**Figure 13 ijms-25-04996-f013:**
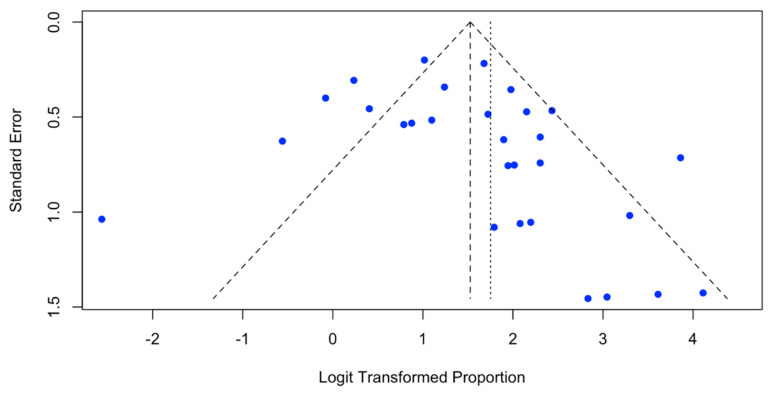
Funnel plot assessment for publication bias in overall response rate (ORR).

**Table 1 ijms-25-04996-t001:** Characteristics of the included studies.

First Author, Year	CAR-T Production Name	Bridging Therapy, Number (%)	Lymphodepletion Regimen	CAR-T Cell Dose	Median Follow-Up (Range), Years or Months
Raje 2019 [29]	Ide-cel	14 (42.4)	fludarabine 30 mg/m^2^ + cyclophosphamide 300 mg/m^2^ daily for 3 days	50–800 × 10^6^ CAR-T cells Dose escalation—50 × 10^6^ (*n* = 3), 150 × 10^6^ (*n* = 6), 450 × 10^6^ (*n* = 9), 800 × 10^6^ (*n* = 3) Dose expansion—150 × 10^6^ (*n* = 2), 450 × 10^6^ (*n* = 10)	11 m (6–23)
Xu 2019 [30]	LCAR-B38M	NR	fludarabine 25 mg/m^2^ + cyclophosphamide 250 mg/m^2^ daily for 3 days, or cyclophosphamide 300 mg/m^2^ daily for 3 days	0.21–1.52 × 10^6^ CAR-T cells/Kg	417 d (12–535)
Cohen 2019 [52]	CART-BCMA	NR	Cohort 1—No LD Cohort 2—Cyclophosphamide 1.5 g/m^2^ Cohort 3—Cyclophosphamide 1.5 g/m^2^	10–500 × 10^6^ CAR-T cells Cohort 1—100–500 × 10^6^ Cohort 2—10–50 × 10^6^ Cohort 3—100–500 × 10^6^	NR
Yan 2020 [31]	CD19 and BCMA CAR-T	NR	fludarabine 30 mg/m^2^ + cyclophosphamide 300 mg/m^2^ daily for 3 days	Anti-BCMA—2.5–6.8 × 10^7^ CAR-T cells/Kg Anti-CD19—1 × 10^7^ CAR-T cells/Kg	20 m
Chen 2020 [32]	CD19 and BCMA CAR-T	NR	fludarabine 30mg/m^2^ daily for 3 days + cyclophosphamide 750mg/m^2^ for 1 day	Anti-BCMA—1–2 × 10^6^ CAR-T cells/Kg Anti-CD19—1 × 10^6^ CAR-T cells/Kg	433 d (230–742)
Wang 2021 [28]	CT103A	1 (5.55)	fludarabine 25 mg/m^2^ + cyclophosphamide 20 mg/Kg daily for 3 days	1–6 × 10^6^ CAR-T cells/Kg 1 × 10^6^ (9), 3 × 10^6^ (6), 6 × 10^6^ (3)	394 d
Cornell 2021 [44]	KITE-585	5 (35.7)	fludarabine 30 mg/m^2^ + cyclophosphamide 300 mg/m^2^ daily for 3 days	3–100 × 10^7^ CAR-T cells	12 m (8.7–14)
Mei 2021 [45]	BM38	NR	fludarabine 25 mg/m^2^ + cyclophosphamide 250 mg/m^2^ daily for 3 days	0.5—4.0 × 10^6^ CAR-T cells/Kg	9 m (0.5–18.5)
Munshi 2021 [46]	Ide-cel	112 (87.5)	fludarabine 30 mg/m^2^ + cyclophosphamide 300 mg/m^2^ daily for 3 days	150–450 × 10^6^ CAR-T cells 150 × 10^6^ (*n* = 4), 300 × 10^6^ (*n* = 70), 450 × 10^6^ (*n* = 54)	13.3 m (0.20–21.2)
Zhang 2022 [33]	BCMA and CD38 CAR-T	NR	fludarabine 30 mg/m^2^ + cyclophosphamide 300 mg/m^2^ daily for 3 days	Anti-BCMA—2.0 × 10^6^ CAR-T cells/Kg Anti-CD38—2.0 × 10^6^ CAR-T cells/Kg	24 m (0.5–33)
Du 2022 [34]	HDS269B	NR	fludarabine 30mg/m^2^ + cyclophosphamide 300mg/m^2^ daily for 3 days	9 × 10^6^ CAR-T cells/Kg	14 m (1–42.5)
Wang 2022 [35]	CD19 and BCMA CAR-T	NR	fludarabine 30mg/m^2^ daily for 3 days + cyclophosphamide 750 mg/m^2^ for 1 day	1 × 10^6^ CAR-T cells/Kg	21.3 m
Zhao 2022 [36]	LCAR-B38M	0	fludarabine 25 mg/m^2^ + cyclophosphamide 250 mg/m^2^, or cyclophosphamide 300 mg/m^2^ daily for 4 days	0.07–2.10 × 10^6^ cells/kg	47.8 m (0.4–60.7)
Mailankody 2022 [37]	MCARH109	16 (94.1)	fludarabine 30 mg/m^2^ + cyclophosphamide 300 mg/m^2^ daily for 3 days	25–450 × 10^6^ CAR-T cells 25 × 10^6^ (*n* = 3), 50 × 10^6^ (*n* = 3), 150 × 10^6^ (*n* = 6), 450 × 10^6^ (*n* = 5)	10.1 m
Qu 2022 [38]	C-CAR088	7 (22.6)	fludarabine 30mg/m^2^ + cyclophosphamide 300mg/m^2^ daily for 3 days	1.0–6.0 × 10^6^ CAR-T cells/Kg 1.0 × 10^6^ (*n* = 4), 3.0 × 10^6^ (*n* = 13), 4.5–6.0 × 10^6^ (*n* = 14)	9.4 m (1.9–24.2)
Tang 2022 [39]	BCMA and CD38 CAR-T	NR	fludarabine 25 mg/m^2^ + cyclophosphamide 250 mg/m^2^ daily for 3 days	0.5–10.0 × 10^6^ CAR-T cells/Kg	11.5 m (6.0–26.0)
Ri 2022 [40]	Cilta-cel	9 (100)	fludarabine 30 mg/m^2^ + cyclophosphamide 300 mg/m^2^ daily for 3 days	0.41–0.72 × 10^6^ CAR-T cells/Kg	8.5 m
Martin 2022 [48]	Cilta-cel	73 (75.3)	fludarabine 30 mg/m^2^ + cyclophosphamide 300 mg/m^2^ daily for 3 days	0.51–0.95 × 10^6^ CAR-T cells/Kg	27.7 m
Mi 2022 [53]	Cilta-cel	NR	fludarabine 30 mg/m^2^ + cyclophosphamide 300 mg/m^2^ daily for 3 days	0.42–0.84 × 10^6^ CAR-T cells/Kg	18 m (0.20–28.0)
Asherie 2022 [51]	HBI0101	3 (15.0)	fludarabine 25 mg/m^2^ + cyclophosphamide 250 mg/m^2^ daily for 3 days	150–800 × 10^6^ CAR-T cells 150 × 10^6^ (*n* = 6), 450 × 10^6^ (*n* = 7), 800 × 10^6^ (*n* = 7)	136 d
Cohen 2023 [41]	Cilta-cel	18 (90.0)	fludarabine 30 mg/m^2^ + cyclophosphamide 300 mg/m^2^ daily for 3 days	0.21–1.11 × 10^6^ CAR-T cells/Kg	11.3 m (0.60–16.0)
Xia 2023 [43]	anti-GPRC5D CAR T	NR	fludarabine 30 mg/m^2^ daily for 3 days + cyclophosphamide 750 mg/m^2^ for 1 day	2 × 10^6^ CAR-T cells/Kg	5.2 m (3.2–8.9)
Mailankody 2023 [47]	ALLO-715	0	fludarabine 90 mg/m^2^ + cyclophosphamide 900 mg/m^2^ daily for 3 days, or cyclophosphamide 900 mg/m^2^ daily for 3 days	40–480 × 10^6^ CAR-T cells 40 × 10^6^ (*n* = 3), 160 × 10^6^ (*n* = 7), 320 × 10^6^ (*n* = 27), 480 × 10^6^ (*n* = 6)	10.2 m (3.8-NE)
Lee 2023 [49]	APRIL CAR-T	4 (36.4)	fludarabine 30 mg/m^2^ + cyclophosphamide 300 mg/m^2^ daily for 3 days	15–900 × 10^6^ CAR-T cells	NR
Oliver-Caldés 2023 [50]	ARI0002h	14 (46.7)	fludarabine 30 mg/m^2^ + cyclophosphamide 300 mg/m^2^ daily for 3 days	0.3–3 × 10^6^ CAR-T cells/Kg	18 m (15–20)
Zhang 2023 [54]	OriCAR-017	2 (20.0)	fludarabine 30 mg/m^2^ + cyclophosphamide 300 mg/m^2^ daily for 3 days	1–6 × 10^6^ CAR-T cells/Kg 1 × 10^6^ (*n* = 3), 3 × 10^6^ (*n* = 4), 6 × 10^6^ (*n* = 3)	238 d (182–307)
Minakata 2023 [42]	Ide-cel	8 (88.9)	fludarabine 30 mg/m^2^ + cyclophosphamide 300 mg/m^2^ daily for 3 days	450 × 10^6^ CAR-T cells	12.9 m (3.30–17.8)
Sanoyan 2023 * [55]	Ide-cel	NR	fludarabine 30 mg/m^2^ + cyclophosphamide 300 mg/m^2^ daily for 3 days	450 × 10^6^ CAR-T cells	5.7 m (0.6–9.0)
Hansen 2023 * [56]	Ide-cel	123 (77.4)	fludarabine (dose adjustment based on creatinine clearance) + cyclophosphamide 300 mg/m^2^ daily for 3 days	NR	6.1m (0.0–13.1)

NR: not reported; NE: Not Estimable; * real-world study.

**Table 2 ijms-25-04996-t002:** Initial characteristics of enrolled participants.

First Author, Year	Extramedullary Disease, Number (%)	High-Risk Cytogenetic Profile, Number (%)	Median No. of Previous Antimyeloma Regimens	Prior Treatment Class	Previous CAR-T Cell Therapy, Number (%)	Previous BCMA Therapy, Number (%)	ASCT before CAR-T, Number (%)
Raje 2019 [29]	9 (27)	15 (46) del(17p), t(4;14), t(14;16)	7 (3–23)	ASCT, proteasome inhibitor, immunomodulatory drug, anti-CD38 mAb	0	0	32 (97)
Xu 2019 [30]	NR	6(35) t(4;14), del(17p)	5 (3–11)	ASCT, proteasome inhibitor, immunomodulatory drug	0	0	8 (47)
Cohen 2019 [52]	7 (28)	24 (96) Defined as del(17p), t(14;16), t(4;14), gain 1q	7 (3–13)	ASCT, proteasome inhibitor, immunomodulatory drug, anti-CD38 mAb	0	0	23 (92)
Yan 2020 [31]	NR	5 (50) t(4;14), 1q21 amp	4 (2–7)	ASCT, proteasome inhibitor, immunomodulatory drug	0	0	6 (60)
Chen 2020 [32]	1 (14)	2 (28) Defined as del(17p), t(4;14) or t(14;16)	5 (2–9)	ASCT, proteasome inhibitor, immunomodulatory drug, anti-CD38 mAb	0	0	2 (28)
Wang 2021 [28]	5 (27.8)	7 (38.9)	4 (3–6)	ASCT, proteasome inhibitor, immunomodulatory drug, anti-CD38 mAb, murine BCMA CART	4 (22.2)	4 (22.2), Murine BCMA CART	6 (33.3)
Cornell 2021 [44]	NR	2 (12)	5.5 (3–8)	ASCT, proteasome inhibitor, immunomodulatory drug, anti-CD38 mAb	0	0	16 (94)
Mei 2021 [45]	9 (39)	17 (74) Includes amplification 1q21, del(17p), del(13q), t(4;14), t(11;14) and t(14;16)	4 (2–9)	ASCT, proteasome inhibitor, immunomodulatory drug	NR	NR	3 (13)
Munshi 2021 [46]	50 (39)	45 (35) Defined as del(17p), t(14;16), t(4;14)	6 (3–16)	ASCT, proteasome inhibitor, immunomodulatory drug, anti-CD38 mAb	0	0	120 (94)
Zhang 2022 [33]	3 (13.6)	19 (86.4) Defined as del(17p), t(14;16), t(4;14	8 (4–12)	Proteasome inhibitor, immunomodulatory drugs, anthracyclines/cyclophosphamide, ASCT	0	0	19 (86.4)
Du 2022 [34]	11 (22.45)	21 (42.86) Defined as del(17p), t(14;16), t(4;14)	4 (2–12)	ASCT, proteasome inhibitor, immunomodulatory drug, anti-CD38 mAb	0	0	14 (28.57)
Wang 2022 [35]	15 (24)	18 (29) Defined as del(17p), t(14;16), t(4;14)	4 (2–17)	ASCT, CAR-T cell infusion, proteasome inhibitor, immunomodulatory drug, anti-CD38 monoclonal antibody	4 (7)	0	17 (27)
Zhao 2022 [36]	22 (29.7)	15 (35.7) Defined as del(17p), t(14;16), t(4;14)	3 (1–9)	ASCT, proteasome inhibitor, immunomodulatory drug, anti-CD38 mAb	0	0	18 (24.3)
Mailankody 2022 [37]	8 (47)	13 (76) Defined as del(17p), t(14;16), t(4;14) and 1q gain	6 (4–14)	ASCT, CAR-T cell infusion, proteasome inhibitor, immunomodulatory drug, anti-CD38 monoclonal antibody, BCMA targeted therapies	8 (47), BCMA CAR-T cell	10 (59)	17 (100)
Qu 2022 [38]	3 (9.7)	15 (48) Defined as del(17p), p53 mutation, t(14;16), t(4;14), t(14;20) and 1q gain	4 (2–13)	ASCT, proteasome inhibitor, immunomodulatory drug, anti-CD38 mAb	NR	NR	7 (22.6)
Tang 2022 [39]	8 (50)	11 (68.8) Including 1q21, del17p	3 (2–3)	ASCT, proteasome inhibitor, immunomodulatory drug	0	0	3 (18.8)
Ri 2022 [40]	3 (33.3)	5 (55.6) Defined as del(17p), t(14;16), t(4;14)	5 (3–7)	ASCT, proteasome inhibitor, immunomodulatory drug, anti-CD38 mAb	0	0	7 (77.8)
Martin 2022 [48]	13 (13)	23 (24) Defined as del(17p), t(14;16), t(4;14)	6 (4–8)	ASCT, proteasome inhibitor, immunomodulatory drug, anti-CD38 mAb	0	0	87 (90)
Mi 2022 [53]	5 (10.4)	21 (43.8) Defined as del(17p), t(14;16), t(4;14)	4 (3–9)	ASCT, proteasome inhibitor, immunomodulatory drug, anti-CD38 mAb	0	0	17 (35.4)
Asherie 2022 [51]	6 (30)	10 (50) Defined as del(17p), t(14;16), t(4;14)	6 (3–13)	ASCT, proteasome inhibitor, immunomodulatory drug, anti-CD38 mAb, anti-BCMA conjugated antibody	NR	9 (45), Anti-BCMA conjugated antibody	17 (85)
Cohen 2023 [41]	5 (25)	3 (15), all del17p	8 (4–13)	ASCT, proteasome inhibitor, immunomodulatory drug, anti-CD38 mAb, noncellular BCMA-directed therapy (ADC or BsAb)	0	20 (100)	20 (100)
Xia 2023 [43]	11 (33)	13 (39) Defined as del(17p), t(14;16), t(4;14) and amp(1q)	4 (2–12)	ASCT, proteasome inhibitor, immunomodulatory drug, anti-CD38 mAb, BCMA CAR-T cell therapy	9 (27), BCMA CAR-T cell	9 (27), BCMA CAR-T cell	6 (18)
Mailankody 2023 [47]	9 (20.9)	16 (37.2) Defined as del(17p), t(14;16), t(4;14)	5 (3–11)	ASCT, proteasome inhibitor, immunomodulatory drug, anti-CD38 mAb, BCMA-directed therapy	0	3 (7)	39 (90.7)
Lee 2023 [49]	3 (27.3)	4 (36.4) Defined as t(4;14), t(14;20) and t(14;16), del(17p), 1q gain, 1p loss	5 (3–6)	ASCT, proteasome inhibitor, immunomodulatory drug, anti-CD38 mAb	0	0	6 (54.5)
Oliver-Caldés 2023 [50]	6 (20)	10 (33) Defined as TP53 alterations, t(14;16), t(4;14)	3.5 (2.8–5.0)	ASCT, proteasome inhibitor, immunomodulatory drug, anti-CD38 mAb	0	0	28 (93)
Zhang 2023 [54]	4 (40)	6 (60) Defined as del(17p), t(14;16), t(4;14)	5.5 (4–10)	ASCT, proteasome inhibitor, immunomodulatory drug, anti-CD38 mAb, BCMA CAR-T cell therapy	5 (50), BCMA CAR-T cell therapy	5 (50), BCMA CAR-T cell therapy	2 (20)
Minakata 2023 [42]	5 (56)	2 (22) Including del(17p), t(4;14)	4 (3–15)	ASCT, proteasome inhibitor, immunomodulatory drug, anti-CD38 mAb	0	0	7 (78)
Sanoyan 2023 [55]	5 (31)	6 (38) Defined as del(17p), t(14;16), t(4;14)	6 (3–12)	ASCT, proteasome inhibitor, immunomodulatory drug, anti-CD38 mAb	NR	NR	16 (100)
Hansen 2023 [56]	76 (48)	49 (35) Defined as del(17p), t(14;16), t(4;14)	7 (4–18)	ASCT, proteasome inhibitor, immunomodulatory drug, anti-CD38 mAb, anti-BCMA therapy	NR	33 (21)	134 (84)

NR: not reported.

**Table 3 ijms-25-04996-t003:** Results of subgroup analysis.

Subgroups	ORR	Grade ≥ 3 CRS
Prior antimyeloma regimens (<5 vs. ≥5)	<5: 89% (95% CI, 84–92) ≥5: 75% (95% CI, 62–85) *p*-value for significance: *p* < 0.01	<5: 13% (95% CI, 8–21) ≥5: 8% (95% CI, 4–14) *p*-value for significance: *p* = 0.18
Prior exposure to BCMA therapy (Yes vs. No)	Yes: 78% (95% CI, 65–87) No: 83% (95% CI, 73–90) *p*-value for significance: *p* = 0.44	Yes: 5% (95% CI, 2–11) No: 13% (95% CI, 8–20) *p*-value for significance: *p* = 0.04
Prior ASCT (<78% vs. ≥78%)	<78%: 87% (95% CI, 80–92) ≥78%: 77% (95% CI, 63–87) *p*-value for significance: *p* = 0.11	<78%: 15% (95% CI, 10–23) ≥78%: 6% (95% CI, 1–11) *p*-value for significance: *p* = 0.02
High-risk cytogenetics (<39% vs. ≥39%)	<39%: 79% (95% CI, 63–89) ≥39%: 85% (95% CI, 77–90) *p*-value for significance: *p* = 0.41	<39%: 7% (95% CI, 4–12) ≥39%: 15% (95% CI, 10–23) *p*-value for significance: *p* = 0.02
ISS stage 3 (<24% vs. ≥24%)	<24%: 86% (95% CI, 75–93) ≥24%: 81% (95% CI, 69–89) *p*-value for significance: *p* = 0.45	<24%: 10% (95% CI, 5–19) ≥24%: 8% (95% CI, 5–13) *p*-value for significance: *p* = 0.61
Extramedullary disease (<28% vs. ≥28%)	< 28%: 85% (95% CI, 72–92) ≥ 28%: 81% (95% CI, 73–87) *p*-value for significance: *p* = 0.52	< 28%: 9% (95% CI, 5–16) ≥ 28%: 10% (95% CI, 5–17) *p*-value for significance: *p* = 0.91
Bridging therapy (<42% vs. ≥42%)	<42%: 75% (95% CI, 44–92) ≥42%: 81% (95% CI, 73–92) *p*-value for significance: *p* = 0.43	<42%: 9% (95% CI, 5–15) ≥42%: 5% (95% CI, 3–7) *p*-value for significance: *p* = 0.07
CAR-T generation (2nd vs. 3rd)	2nd: 83% (95% CI, 75–88) 3rd: 80% (95% CI, 48–95) *p*-value for significance: *p* = 0.86	2nd: 10% (95% CI, 6–15) 3rd: 10% (95% CI, 5–18) *p*-value for significance: *p* = 0.93
Upper infusion threshold (<490 × 10^6^ cells or 2.05 ×10^6^ cells/kg vs. ≥490 × 10^6^ cells or 2.05 ×10^6^ cells/kg)	<490 × 10^6^ cells or 2.05 ×10^6^ cells/kg: 84% (95% CI, 74–90) ≥490 × 10^6^ cells or 2.05 ×10^6^ cells/kg: 81% (95% CI, 66–90) *p*-value for significance: *p* = 0.71	<490 × 10^6^ cells or 2.05 ×10^6^ cells/kg: 9% (95% CI, 5–16) ≥490 × 10^6^ cells or 2.05 ×10^6^ cells/kg: 12% (95% CI, 7–19) *p*-value for significance: *p* = 0.41

## Data Availability

The datasets generated and/or analyzed during the current study are available from the corresponding author on reasonable request, provided no ethical, legal, or privacy issues are raised.

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
