# Peer review of "An Assessment of the Effectiveness and Safety of Chimeric Antigen Receptor T-Cell Therapy in Multiple Myeloma Patients with Relapsed or Refractory Disease: A Systematic Review and Meta-Analysis"

_ijms, 2024, doi:10.3390/ijms25094996_

Round 1

Reviewer 1 Report

Comments and Suggestions for Authors

In the manuscript entitled "Assessment of the effectiveness and safety of CAR-T cell therapy in multiple myeloma patients with relapsed or refractory disease: a systematic review and meta-analysis", Pereira and Bergantim have assembled an informative and comprehensive meta-analysis of clinical trials reported to date for the use of BCMA-targeted CAR-T therapies for the treatment of multiple myeloma. Overall, the work is found to be well written, insightful, and scientifically impactful. 

There are some criticisms that should be addressed:

(1) Page 2: "The CAR structure comprises an external component featuring a single-chain variable fragment targeting the desired antigen" - While scFvs are the most common, the binding domain of the CAR can be derived from various receptors or other binding elements. This detail is particularly important because BCMA-targeting ciltabtagene autoleucel uses a dual-nanobody structure, which is unique among the FDA approved therapies. 

(2) In Table 1 it would be helpful to include a reference to the name of the product tested in the various trials. As many of them may be either ide cel or cilta cel, it would be helpful to identify this.

(3) Also in Table 1, it would be helpful if authors can provide information on the specific designs used for the CARs (binder, spacer, transmembrane, and signaling domains; eg Nanobody-CD8h-CD8TM-BBz), though it is understandable if this data is not readily available to the authors. 

(4) Page 15 - In the analysis of DOR it is not clear why only 5 studies were included. Authors should give some explanation here

(5) Page 15-17 - For CRS, neurotox, hematological malignancy, and infections authors define severe as > grade 3 but it is not clear what grading scheme is used. Presumably most reports use the same grading scheme? If authors can provide  a reference to the grading scheme used for these it would be helpful to the manuscript. Furthermore, if authors can discuss briefly whether a common grading scheme was used in all reported studies, it would be helpful in interpretation of this data. 

(6) Page 17 - "prior exposure to BCMA therapy (< yes vs. ≥ no)" - I assume this should be simply (yes or no)

(7) Page 18 - Authors are commended for their interesting work on subgroup analysis. Have authors investigated whether CAR-T dose correlates with response. Given that dose is reported in Table 1, it was curious that it was not analyzed as a metric here. If authors add this, they should also provide some discussion of the result in the discussion section. 

(8)  Page 23 - Authors provide a nice discussion on alterative treatment regimens but do not put it in context of their own findings. Authors are encouraged to shorten this discussion and provide some context from this study. If no analysis of such alternative dosing (or dosing in general) is possible or useful in this meta-analysis, authors should explicitly state so and provide a reason (ie lack of data). 

(9) Page 23 - Authors discuss whether previous lines of BCMA therapy have an effect on response to BCMA CAR-T but do not mention their own findings. This should be added here. Despite some disagreement in regards to the significance of prior BCMA treatment as a predictor of response, overall the data seems to indicate that the magnitude of the effect is relatively small. 

(10) Conclusions - The paper should include a clear and concise summary of the findings. Authors should restate the general findings of the meta analysis including ORR, MRD-neg, DOR, and toxicity observations. Also the lack of obvious subgroup correlations with the exception of number of prior lines of therapy is interesting and could be included in conclusions. Authors should also consider including this in the abstract. 

Reviewer 2 Report

Comments and Suggestions for Authors

'The manuscript provides a meta-analysis of reports of CAR T cell therapy for multiple myeloma and provides a review on the state of the field.

My minor comments:

In section 2.1, it seems to imply that all the studies used an independent review committee to assess response. My understanding was that such committees are typically only used for late-stage (phase 3) studies. Please confirm the text or revise.

In Table 1, please indicate which of the 2 studies were real-world studies.

In section 3.2, it is indicated that 3 studies were multicenter. That seems low to me. Do you mean that 3 studies enrolled patients in more than one country (rather than "multicenter" referring to multiple sites within one country)?
